# Effect of Different Processing Methods on the Chemical Constituents of Scrophulariae Radix as Revealed by 2D NMR-Based Metabolomics

**DOI:** 10.3390/molecules27154687

**Published:** 2022-07-22

**Authors:** Xiaohui Duan, Mina Zhang, Huan Du, Xiu Gu, Caihong Bai, Liuqiang Zhang, Kaixian Chen, Kaifeng Hu, Yiming Li

**Affiliations:** 1School of Pharmacy, Shanghai University of Traditional Chinese Medicine, 1200 Cailun Road, Shanghai 201203, China; duanxiaohui2019@163.com (X.D.); mina1215737003@163.com (M.Z.); 04100217@163.com (L.Z.); kxchen@simm.ac.cn (K.C.); 2Innovative Institute of Chinese Medicine and Pharmacy, Chengdu University of Traditional Chinese Medicine, 1166 Liutai Avenue, Chengdu 611137, China; 3School of Pharmacy, Chengdu University of Traditional Chinese Medicine, Chengdu 611137, China; dave911dh@163.com (H.D.); guxiu@stu.cdutcm.edu.cn (X.G.); baicaihong@stu.cdutcm.edu.cn (C.B.)

**Keywords:** 2D NMR, Scrophulariae Radix, “sweating” processing, processing methods, chemical markers

## Abstract

Scrophulariae Radix (SR) is one of the oldest and most frequently used Chinese herbs for oriental medicine in China. Before clinical use, the SR should be processed using different methods after harvest, such as steaming, “sweating”, and traditional fire-drying. In order to investigate the difference in chemical constituents using different processing methods, the two-dimensional (2D) ^1^H-^13^C heteronuclear single quantum correlation (^1^H-^13^C HSQC)-based metabolomics approach was applied to extensively characterize the difference in the chemical components in the extracts of SR processed using different processing methods. In total, 20 compounds were identified as potential chemical markers that changed significantly with different steaming durations. Seven compounds can be used as potential chemical markers to differentiate processing by sweating, hot-air drying, and steaming for 4 h. These findings could elucidate the change of chemical constituents of the processed SR and provide a guide for the processing. In addition, our protocol may represent a general approach to characterizing chemical compounds of traditional Chinese medicine (TCM) and therefore might be considered as a promising approach to exploring the scientific basis of traditional processing of TCM.

## 1. Introduction

Scrophulariae Radix (SR), the roots of *Scrophularia ningpoensis* Hemsl. (Scrophulariaceae), was first recorded in the earliest Chinese medicinal classic, Shennong’s Materia Medica [1,2,3]. It has been used to treat rheumatism, pharyngalgia, arthritis, neuritis, constipation, conjunctival congestion, and diabetes [4,5,6,7] due to its excellent traditional therapeutic effects on “cooling blood”, “nourishing yin”, and “purging fire” for removing toxins [1,8]. In the last decades, SR has been extensively investigated and shown to contain iridoid glycosides, phenylpropanoid glycosides, organic acids, and triterpenes [1,8,9]. Pharmacological studies and clinical practice have demonstrated that iridoid glycosides and phenylpropanoid glycosides possess various bioactivities, such as anti-inflammatory, antimicrobial, antitumor, antioxidant, and antiviral activities [9,10,11,12,13]. Harpagide and harpagoside are the quality markers recorded in the Chinese Pharmacopoeia 2020 [14].

In TCM, processing is a common practice, aiming at enhancing efficacy and reducing toxicity in clinical application [15,16,17]. According to the Chinese Pharmacopoeia (2020), SR needs to be processed by “sweating” [14]. Sweating processing is one of the common methods in TCM raw material processing. The detailed procedure for SR is as follows. The raw material is baked with a low fire to semi-dry and then piled up to sweat to make the internal moisture spillover. The procedure is repeated several times until the material is dry. The sweating process makes the SR become soft and change color to increase fragrance and reduce irritation [18]. There are two other important methods for SR processing recorded in ancient classical TCM processing books such as Lei Gong’s Moxibustion Theory and the Compendium of Materia Medica, using steaming and hot-air drying methods [19]. Although there are two articles describing the chemical constituents in SR after different processing methods [8,10], they only investigated the change of a limited number of known components in SR. However, for traditional Chinese medicine with complex chemical components, the information obtained from a few known chemical components is not comprehensive enough to clarify the impact of traditional processing on SR. Therefore, it is essential to develop a method to comprehensively characterize compounds in SR and its processed products.

Untargeted metabolomics is a technology allowing an unbiased and hypothesis-free assessment of the metabolome [20]. Nuclear magnetic resonance (NMR) is a powerful platform in untargeted metabolomics due to its ability to analyze essentially all types of known and unknown natural products [21]. Recently, two-dimensional (2D) NMR has attracted attention because it can provide more detailed metabolite fingerprints [22,23]. The 2D NMR experiment commonly used in metabolomics is 2D ^1^H-^13^C heteronuclear single quantum coherence (HSQC) [24]. The large ^13^C chemical shift dispersion in the 2D ^1^H-^13^C HSQC spectra allowed us to analyze a wide range of metabolites without prior separation [25], and the information provided by specific atomic correlations can assist in the identification and assignment of known metabolites as well as illuminate the NMR information of unknown metabolites [26]. Based on the rather inclusive and publicly available databases (HMDB, BMRB), multiple well-resolved ^1^H-^13^C cross-peaks in 2D ^1^H-^13^C HSQC spectra have been frequently used for metabolite identification. A few works have studied the characteristic metabolic changes of the crust from dry-aged beef and metabolites of six different laboratory *Escherichia coli* strains by using this method [27,28]. However, few studies have used ^1^H-^13^C HSQC spectra in untargeted metabolomics analysis of TCM.

In our current study, 2D ^1^H-^13^C HSQC-based metabolomics was used to extensively characterize the chemical components in SR processed by freeze-drying, steaming for different durations (S), hot-air drying (HD), and “sweating” (SW). Firstly, the extraction conditions of SR metabolomics samples were optimized to fully extract metabolites and save data acquisition time. Secondly, we developed two in-house TopSpin AU programs to automate the batch processing of 1H NMR and an in-house NMRPipe script to automate the processing of all 2D 1H-13C HSQC spectra in-batch. Finally, the differential features selected were analyzed using NPid and preliminarily assigned by comparing with the customized NMR database of SR obtained through literature data mining and public spectral databases (HMDB, BMRB). This study provides a useful and quick strategy to comprehensively analyze chemical compounds of SR with the 2D ^1^H-^13^C HSQC features and therefore might be considered as a promising approach for exploring the scientific basis in traditional processing of SR.

## 2. Results and Discussion

### 2.1. Samples Collection and Processing

A total of 99 samples were obtained by processing fresh roots of SR with four different methods (FD, S, SW, HD). The detailed information on sample processing is shown in Appendix A.

### 2.2. Optimization of Extraction Conditions

#### 2.2.1. Selecting the Optimum Extraction Solvent

Dried samples of SR (250 mg) were exactly weighed and extracted in parallel with five different solvents, namely 20%, 40%, 60%, 80%, 100% *v*/*v* methanol/water solutions. The 1D ^1^H NMR spectra of the five extracts were obtained. Segmented absolute integrals of three regions were calculated except for the signals from residual water and protonated solvents, including the aromatic region (10.00–5.60 ppm), the polar (carbohydrates) region (5.60–3.33 ppm), and the aliphatic region (3.30–0.05 ppm) (Appendix A). The absolute segmental integrals of the extracts with five different extraction solvents were compared to evaluate the extraction efficiency of the SR metabolites. The 60% methanol/water solutions could extract more aromatic compounds and carbohydrates, whereas 80% methanol/water solutions could extract more aliphatic compounds. However, 60% methanol/water solutions could extract more global metabolites. Finally, 60% methanol/water solutions were selected as the optimum extraction solvents.

#### 2.2.2. Optimization of the Solid/Liquid Ratio of the Extraction

The effect of solid/liquid ratio on the extraction yield was investigated with a SR samples vs. different volumes of 60% methanol/water solutions at 20, 60, 100, 140, 180, 200 mL/g. With the increasing volume of solvent, the total integrals of ^1^H NMR spectra increased gradually and reached the plateau when the solvent to solid ratio was 140 mL/g (Appendix A), indicating that the volume of solvents used was enough to completely extract the SR metabolites. In practice, considering the possible variation of the quantity of metabolites among different SR samples, the optimum solvent to solid ratio was appropriately relaxed to 160 mL/g to ensure complete extraction of SR metabolites.

#### 2.2.3. Optimum Sample Amount Used per NMR Sample

The use of excessive SR per NMR sample may lead to oversaturation of the extracts in the NMR deuterated solvents, which may bring about non-linear response or even insensitivity of the intensity of NMR signals to the real quantity of metabolites in the original SR sample. In the aromatic region, the range of SR sample amount of 100 to 500 mg was located in the linear regime. In the sugar region, the range of SR sample amount of 100 to 300 mg was located in the linear regime. In the aliphatic region, the range of SR sample amount of 100 to 500 mg was located in the linear regime (Appendix A). The integral of 1D ^1^H NMR spectra showed that the optimum amount of SR used per NMR sample was 250 mg, and the concentration of the NMR samples made the acquisition time of ^1^H-^13^C HSQC spectra reasonably short (1 h 52 min).

### 2.3. Comparison of SR Being Steamed Processing for Different Time

#### 2.3.1. Multivariate Statistical Analysis

In order to study the effects of different steaming durations on the metabolites of SR, PCA and OPLS-DA were performed on FD and steamed samples (S01, S02, S04, S08, S12, S24, S48, S72). Figure 1 showed that the difference between steamed and FD samples became more and more significant with the extension of steaming time. That is, the longer the steaming time, the more significant the change of chemical composition of SR. Comparison of FD and S01 in the PCA score plot of 2D ^1^H-^13^C HSQC spectra showed no significant separation between FD and S01 (Figure 1A), whereas FD and S02 were more dispersed (Figure 1B). Comparison of FD and S04-S72 (Figure 1C–H) showed a gradually separation of the two groups.

In the supervised OPLS-DA score plots (Figure 2A–H), clear discrimination was observed between FD and steamed samples (S01, S02, S04, S08, S12, S24, S48, S72). All OPLS-DA models are valid with R^2^(cum) and Q^2^(cum) values exceeding 0.8 (Appendix A). Variables with VIP > 1.1 and |*p*(corr)| > 0.9 were selected as significantly differential features and highlighted in magenta in the volcano plot (Figure 2a–h). Comparison between FD and steamed samples (S01, S02, S04, S08, S12, S24, S48, S72), with total 215 features (10, 10, 20, 28, 29, 61, 123, 150 for samples at each steaming time, respectively, calculate the intersection of eight groups), showed significantly differential features (Table 1). Univariate statistical analysis was performed to further confirm that the selected features were statistically significant with *p* value < 0.01 using Student’s *t*-test.

#### 2.3.2. Change of Chemical Constituents of SR Steamed for Different Durations

Out of the 215 differential features, 131 features were assigned to 20 compounds (Table 1). The identified differential metabolites were further confirmed by overlaying and comparing the 2D ^1^H-^13^C HSQC spectra of the mixture with the standards of the candidate metabolites. The boxplots of the intensities of 20 differential metabolites in SR steamed for different durations are shown in Figure 3.

Taking FD samples as a reference, the relative contents of 15 differential metabolites, namely harpagoside, cinnamic acid, harpagide, aucubin, 6-*O*-methyl-catalpol, amino acids (leucine, threonine, tyrosine, methionine, valine, arginine, isoleucine), and oligosaccharides (sucrose, raffinose, and stachyose), decreased significantly (*p* value < 0.01) at varying steaming duration, while the relative contents of three differential monosaccharides, namely β-glucose, fructose, and glucuronic acid, increased with the steaming time. The relative contents of glutamine and 4-aminobutyric acid increased at the beginning (from 1 to 24 h and from 1 to 2 h, respectively) and then gradually decreased with steaming time.

With the extension of steaming time, the decrease in oligosaccharides and glycosides and increase in monosaccharides indicated possible hydrolysis of the oligosaccharides and glycosides during the steaming. We speculate that the decrease in the amino acids with the extension of steaming could be related to thermally induced Maillard reaction [29], which might be largely responsible for the browning sesames of the processed SR [30]. The boxplots of the intensities of the other 84 unidentified differential spectral features are shown in Appendix A. Further phytochemical study on the steamed SR might be helpful to determine their chemical identities.

The majority of chemical constituents of SR showed significant change after being steamed for 4 h, while the active components iridoids did not decrease significantly. Therefore, S04 samples were selected to represent samples processed by steaming and compared with SW and HD samples.

**Table 1 molecules-27-04687-t001:** Detailed information and assignments of the differential features (F) among samples of SR steamed for different durations with VIP > 1.1 and |*p*(corr)| > 0.9 by OPLS-DA and *p* value < 0.01 using univariate Student’s *t*-test.

Feature	*δ* _H_	*δ* _C_	Assignment	Feature	*δ* _H_	*δ* _C_	Assignment	Feature	*δ* _H_	*δ* _C_	Assignment
F115	2.72	46.66	aucubin	F121	3.02	49.38	aucubin	F124	5.05	99.01	aucubin
F128	5.81	131.75	aucubin	F127	5.12	107.85	aucubin	F135	6.32	142.93	aucubin
F179	4.33	62.72	aucubin	F196	4.35	62.72	aucubin	F283	3.84	79.19	harpagoside
F299	5.01	108.14	harpagoside	F346	2.95	56.61	harpagoside	F372	6.16	96.43	harpagoside
F408	6.47	145.71	harpagoside	F146	1.52	24.36	harpagoside	F558	2.29	47.69	harpagoside
F581	2.04	47.68	harpagoside	F773	6.55	121.43	harpagoside, cinnamic acid	F225	7.63	131.07	harpagoside, cinnamic acid
F679	7.69	148.31	harpagoside, cinnamic acid	F101	4.72	101.44	aucubin, harpagoside	F118	1.26	26.82	harpagide
F242	5.03	109.70	harpagide	F216	3.78	79.47	harpagide	F219	4.67	101.09	harpagide
F279	6.37	144.22	harpagide	F254	2.57	60.43	harpagide	F273	5.73	95.26	harpagide
F375	1.95	48.66	harpagide	F294	1.85	48.66	harpagide	F607	1.83	48.66	harpagide
F495	1.97	48.66	harpagide	F106	3.30	76.01	aucubin, harpagoside, harpagide	F268	3.31	75.64	aucubin, harpagoside, harpagide
F244	3.35	72.89	aucubin, harpagoside, harpagide, glucose	F220	3.77	61.50	6-*O*-methyl-catalpol	F218	2.35	38.86	6-*O*-methyl-catalpol
F229	6.40	143.71	6-*O*-methyl-catalpol	F207	5.08	105.85	6-*O*-methyl-catalpol	F520	4.18	63.00	6-*O*-methyl-catalpol
F138	4.82	101.35	6-*O*-methyl-catalpol	F201	5.05	97.23	6-*O*-methyl-catalpol	F123	4.20	62.79	6-*O*-methyl-catalpol
F184	3.35	76.10	6-*O*-methyl-catalpol	F194	2.60	44.63	6-*O*-methyl-catalpol	F178	4.23	62.72	6-*O*-methyl-catalpol
F111	3.48	78.90	aucubin, harpagoside, harpagide, 6-*O*-methyl-catalpol, glucuronic acid	F107	3.38	79.20	aucubin, harpagoside, harpagide, 6-*O*-methyl-catalpol, glucose	F108	3.37	79.39	aucubin, harpagoside, harpagide, 6-*O*-methyl-catalpol
F42	3.45	79.22	aucubin, harpagoside, harpagide, 6-*O*-methyl-catalpol, glucuronic acid	F300	3.41	79.25	aucubin, harpagoside, harpagide, 6-*O*-methyl-catalpol	F228	3.41	79.45	aucubin, harpagoside, harpagide, 6-*O*-methyl-catalpol, glucose
F649	1.53	27.47	isoleucine	F367	3.39	78.91	aucubin, harpagoside, harpagide, 6-*O*-methyl-catalpol, glucose	F460	3.61	62.61	isoleucine
F238	0.96	14.06	isoleucine	F611	1.98	39.00	isoleucine	F224	1.03	17.53	isoleucine
F159	7.18	133.57	tyrosine	F330	2.65	32.01	methionine	F1423	3.00	38.63	tyrosine
F659	3.20	38.64	tyrosine	F147	6.85	118.65	tyrosine	F248	3.88	59.18	tyrosine
F339	2.05	28.38	glutamine	F357	3.01	38.65	tyrosine	F389	3.22	38.65	tyrosine
F186	3.53	63.50	threonine, valine	F261	2.09	28.39	glutamine	F145	2.07	28.41	glutamine
F596	1.76	43.04	leucine	F95	1.35	22.64	threonine	F235	0.99	25.00	leucine
F297	1.73	27.11	leucine, arginine	F230	0.97	23.83	leucine	F637	1.66	43.03	leucine
F198	1.94	30.71	arginine	F350	1.67	27.09	arginine	F470	3.21	43.69	arginine
F40	3.83	75.61	sucrose	F213	1.90	30.67	arginine	F622	3.82	62.76	sucrose
F31	3.80	63.37	sucrose	F53	3.44	72.46	sucrose	F28	3.83	65.34	sucrose
F19	3.76	75.90	sucrose	F55	3.78	63.39	sucrose	F18	3.54	74.19	sucrose
F5	3.81	65.32	sucrose, stachyose, raffinose	F2	3.66	64.87	sucrose, stachyose	F12	3.78	65.34	sucrose, stachyose, raffinose
F11	5.43	95.10	sucrose, stachyose, raffinose	F6	4.18	79.84	sucrose, stachyose, raffinose	F9	3.85	84.66	sucrose, stachyose, raffinose
F23	4.95	101.65	stachyose, raffinose	F65	5.41	95.23	sucrose, stachyose, raffinose	F20	4.12	71.75	stachyose
F25	3.67	69.01	stachyose, raffinose	F22	4.02	69.01	stachyose, raffinose	F21	4.06	74.43	stachyose, raffinose
F46	3.23	77.36	glucose, glucuronic acid	F56	3.71	76.20	sucrose, glucose, glucuronic acid	F51	5.21	95.42	glucuronic acid
F258	3.25	77.38	glucose	F116	3.81	74.47	glucose	F54	3.48	74.80	glucose
F137	3.92	63.97	glucose	F103	4.58	99.25	β-glucose	F102	3.21	77.44	glucose
F97	4.01	66.26	fructose	F155	3.46	74.92	glucose	F57	4.03	66.26	fructose
F328	3.70	67.08	fructose	F44	3.81	70.65	fructose	F180	3.98	79.17	fructose
F41	3.71	67.08	fructose	F74	3.70	66.25	fructose	F154	4.07	85.43	fructose
F61	3.55	65.84	fructose	F35	3.53	67.08	fructose	F321	3.51	67.08	fructose
F48	3.68	66.24	fructose	F68	4.08	78.69	fructose	F163	3.79	70.65	fructose
F71	3.53	65.82	fructose	F692	3.02	41.93	4-aminobutyric acid	F199	2.32	37.41	4-aminobutyric acid
F175	3.01	42.37	4-aminobutyric acid	F192	1.91	26.53	4-aminobutyric acid	F624	2.29	32.19	valine
F237	1.06	20.98	valine	F232	1.02	19.63	valine	F223	3.92	79.83	
F260	3.24	56.82		F386	1.94	25.75		F182	3.21	56.76	
F546	4.05	58.63		F758	2.67	39.47		F92	4.14	61.30	
F34	4.62	99.40		F946	2.83	39.45		F975	3.12	44.34	
F791	3.81	60.65		F52	3.73	68.65		F503	3.51	70.54	
F473	3.86	55.36		F740	6.78	118.64		F864	2.81	39.47	
F292	2.50	28.39		F156	2.48	28.41		F309	2.41	36.86	
F872	6.67	123.63		F176	2.40	32.70		F853	6.79	119.26	
F90	2.38	32.74		F181	4.02	84.29		F204	2.36	32.69	
F285	2.46	28.39		F278	5.04	110.79		F642	3.59	49.68	
F708	2.11	32.99		F288	3.59	72.65		F49	3.98	68.66	
F36	3.85	72.91		F315	5.11	110.38		F120	3.50	74.41	
F487	1.30	19.68		F322	4.00	86.92		F500	3.67	56.35	
F502	4.18	85.47		F354	4.24	82.63		F545	2.82	37.70	
F505	3.82	55.44		F480	5.06	110.70		F772	3.40	58.90	
F636	2.04	24.55		F557	4.25	84.47		F1400	5.43	75.92	
F405	3.25	77.58		F608	3.96	80.14		F1430	4.97	72.81	
F699	2.21	32.98		F616	4.10	84.97		F1483	5.43	74.44	
F447	2.75	39.54		F645	4.09	84.02		F117	3.51	60.15	
F43	3.59	77.32		F646	3.91	76.55		F188	3.76	74.20	
F723	3.42	57.74		F777	1.79	29.00		F195	3.75	89.84	
F66	3.80	84.11		F921	2.15	23.05		F739	6.56	138.61	
F72	4.00	73.00		F745	1.91	33.03		F287	4.18	77.06	
F89	4.08	77.78		F1209	5.90	128.65		F457	4.06	84.65	
F349	2.14	16.84		F13	4.06	77.06		F29	3.83	71.78	
F32	4.16	71.74		F125	4.48	84.00		F206	4.06	79.80	
F174	3.87	73.28		F422	3.56	63.35		F750	5.43	85.29	
F193	3.61	66.25		F845	4.29	84.69		F684	3.36	49.67	
F256	4.10	84.45		F932	4.16	85.44					

### 2.4. Comparison of SR Processed by SW, HD, and S04

#### 2.4.1. Multivariate Statistical Analysis

In order to study the effect of different processing methods on chemical constituents of SR, PCA and PLS-DA were performed on the samples processed by SW, HD, and S04. PCA gave a model with five principal components (PCs), which cumulatively accounted for 74.7% of the total variance. In the PCA score plot, HD and S04 overlapped with each other, suggesting that these were relatively similar in their compositional patterns in 2D ^1^H-^13^C HSQC spectra. Compared with HD and S04, SW showed significant metabolic differences in the PCA model (Figure 4A). PLS-DA generated a two-dimensional model with goodness of fit R^2^(cum) of 0.934 and predictive power Q^2^(cum) of 0.901. Further, the permutation test checked the validity and the degree of overfit for the PLS models. As shown in Appendix A, the intercept (R^2^ = 0.209, Q^2^ = −0.23) in the permutation plot is a measure of the overfit, which suggests that the models are reliable and not overfitted. The LV1 and LV2 components of the model explained 42.2% and 10.7% of the total variance, respectively. The differently processed samples were clearly divided into three groups in the score plot of PLS-DA (Figure 4B). A total of 81 features were selected as preliminary differential variables based on their VIP (>1.0) and |*p*(corr) | (>0.8) values as shown in the volcano plot (Figure 4C,D). Univariate statistical analysis was performed to further confirm that the selected features (Table 2) were statistically significant with *p* value < 0.01 according to ANOVA.

#### 2.4.2. Identification of Differential Metabolites of SW, HD, and S04

The effects of different processing methods on chemical constituents of SR can be characterized using 81 differential features, of which 49 features were assigned to seven compounds (Table 2). The boxplots of the intensities of these seven differential metabolites are shown in Figure 5. Compared with SR processed by HD and S04, the relative content of six differential metabolites, harpagoside, cinnamic acid, 6-*O*-methyl-catalpol, aucubin, methionine, and tyrosine, in SR processed by SW was generally lower, while the relative content of α/*β*-glucose was higher in SR processed by SW.

The boxplots of the intensities of the other 32 unidentified differential features in ^1^H-^13^C HSQC spectrum are shown in Appendix A. Among these 32 unidentified features, 16 of them were characteristic of SR processed by SW, while three of them did not show in SR processed by SW. Further phytochemical study to elucidate the chemical identities of these unidentified features is necessary and of significance to deepening our understanding of the effects of different processing methods on the chemical constituents of SR.

**Table 2 molecules-27-04687-t002:** Detailed information and assignments of the differential features (F) among differently processed SR with VIP > 1.0 and |*p*(corr)| > 0.8 by PLS-DA and *p* value < 0.01 by univariate ANOVA.

Feature	*δ* _H_	*δ* _C_	Assignment	Feature	*δ* _H_	*δ* _C_	Assignment	Feature	*δ* _H_	*δ* _C_	Assignment
F115	2.72	46.66	aucubin	F121	3.02	49.38	aucubin	F124	5.05	99.01	aucubin
F135	6.32	142.93	aucubin	F128	5.81	131.75	aucubin	F179	4.33	62.72	aucubin
F127	5.12	107.85	aucubin	F685	2.27	47.68	harpagoside	F299	5.01	108.14	harpagoside
F408	6.47	145.71	harpagoside	F372	6.16	96.43	harpagoside	F346	2.95	56.61	harpagoside
F146	1.52	24.36	harpagoside	F625	2.06	47.68	harpagoside	F283	3.84	79.19	harpagoside
F712	7.66	148.31	harpagoside, cinnamic acid	F225	7.63	131.07	harpagoside, cinnamic acid	F773	6.55	121.43	harpagoside, cinnamic acid
F671	6.53	121.43	harpagoside, cinnamic acid	F101	4.72	101.44	aucubin, harpagoside	F244	3.35	72.89	aucubin, harpagoside, glucose
F138	4.82	101.35	6-*O*-methyl-catalpol	F207	5.08	105.85	6-*O*-methyl-catalpol	F194	2.60	44.63	6-*O*-methyl-catalpol
F123	4.20	62.79	6-*O*-methyl-catalpol	F184	3.35	76.10	6-*O*-methyl-catalpol	F229	6.40	143.71	6-*O*-methyl-catalpol
F201	5.05	97.23	6-*O*-methyl-catalpol	F218	2.35	38.86	6-*O*-methyl-catalpol	F220	3.77	61.50	6-*O*-methyl-catalpol
F42	3.45	79.22	aucubin, harpagoside, 6-*O*-methyl-catalpol, glucose	F333	3.39	79.72	aucubin, harpagoside, 6-*O*-methyl-catalpol, glucose	F111	3.48	78.90	aucubin, harpagoside, 6-*O*-methyl-catalpol
F108	3.37	79.39	aucubin, harpagoside, 6-*O*-methyl-catalpol	F367	3.39	78.91	aucubin, harpagoside, 6-*O*-methyl-catalpol, glucose	F130	3.46	79.14	aucubin, harpagoside, 6-*O*-methyl-catalpol
F107	3.38	79.20	aucubin, harpagoside, 6-*O*-methyl-catalpol, glucose	F211	3.44	79.30	aucubin, harpagoside, 6-*O*-methyl-catalpol	F494	3.80	56.89	methionine
F699	2.21	32.98	methionine	F349	2.14	16.84	methionine	F330	2.65	32.01	methionine
F708	2.11	32.99	methionine	F659	3.20	38.64	tyrosine	F1423	3.00	38.63	tyrosine
F248	3.88	59.18	tyrosine	F103	4.58	99.25	β-glucose	F150	5.19	95.24	α-glucose
F102	3.21	77.44	glucose	F50	3.73	57.21		F1209	5.90	128.65	
F88	3.98	72.38		F667	5.14	98.33		F125	4.48	84.00	
F77	3.91	74.04		F1055	1.58	30.33		F340	1.24	17.22	
F117	3.51	60.15		F195	3.75	89.84		F514	1.94	24.72	
F622	3.82	62.76		F47	1.35	23.18		F610	2.36	37.79	
F723	3.42	57.74		F606	4.01	82.57		F631	4.45	105.80	
F988	1.93	29.66		F689	7.38	132.36		F741	3.13	51.58	
F717	1.95	29.64		F727	1.99	29.63		F760	3.35	76.83	
F761	3.26	55.60		F772	3.40	58.90		F989	4.81	101.63	
F777	1.79	29.00		F794	3.35	55.66		F800	4.02	66.89	
F182	3.21	56.76		F503	3.51	70.54		F546	4.05	58.63	

## 3. Materials and Methods

### 3.1. Chemicals and Reagents

Methanol-*d*_4_ (99.8 atom% D) was purchased from Energy Chemical (Shanghai, China). Deuterium oxide (99.8 atom% D) was obtained from Cambridge Isotope Laboratories, Inc. (Andover, MA, USA). The 2, 2, 3, 3-*d*_4_- 3-(trimethylsilyl) propionic acid sodium salt (TSP, 98 atom% D) was bought from Alfa Aesar (Andover, MA, USA). Natural products, namely aucubin (7683), harpagoside (5314), harpagide (5239), and angroside C (5471), were purchased from Shanghai Standard Technology Co., Ltd. (Shanghai, China). Cinnamic acid (19121235) was obtained from Shanghai Tauto Biotech Co., Ltd. (Shanghai, China). Tryptophan (6135), methionine, and tyrosine (140609-201914) were bought from the China Academy of Food and Drug Control (Beijing, China). Catalpol and glutamine were purchased from Sichuan Weikeqi Biological Technology Co. Ltd. (Chengdu, China). The 6-*O*-methyl-catalpol was made by the research group. The purity of each standard compound was over 98%.

### 3.2. Samples Collection and Processing

#### 3.2.1. Fresh SR Raw Material Collection

Biological triplicate samples of the roots of *Scrophularia ningpoensis* were collected from each of three different locations, Panfengxiang (PPS1-3), Dapanzhen (PDS1-3), and Fangqianzhen (PFS1-3), in Pan’an County in Zhejiang Province, China, in November 2020. The botanical origin of the materials was identified by Professor Yiming Li (School of Pharmacy, Shanghai University of Traditional Chinese Medicine, China.), and voucher specimens (no. 2020SRP, no. 2020SRD, no. 2020SRF) were deposited at the School of Pharmacy, Shanghai University of Traditional Chinese Medicine, China.

#### 3.2.2. SR Raw Material Samples Processing

Samples went through processing with different methods: vacuum freeze-drying, steaming (S) for different durations, sweating (SW), and hot-air drying (HD). The FD samples (*n* = 9) were first frozen in an ultra-low temperature refrigerator at −80 °C and then placed in a freeze dryer with a cold-trap temperature of −80 ºC to dry to constant weight. Since the steaming time was extremely important to the processing, the steamed samples were divided into different subgroups according to the steaming time for 1 (S01, *n* = 9), 2 (S02, *n* = 9), 4 (S04, *n* = 9), 8 (S08, *n* = 9), 12 (S12, *n* = 9), 24 (S24, *n* = 9), 48 (S48, *n* = 9), and 72 h (S72, *n* = 9), and all steamed samples were subsequently dried to constant weight at 55 °C in the oven. The SW samples (*n* = 9) were processed as follows: dried for 24 h at 55 °C in the oven, placed into a bag for 48h as the first sweating then removed from the bag, dried for 24 h at 55 °C in the oven, placed into a bag for 48 h as the second sweating, and finally removed from the bag and dried to constant weight at 55 °C in the oven. The HD samples (*n* = 9) were processed with oven drying at 55 °C. All processed samples were then ground into rough powder and kept in a refrigerator at 4 °C for subsequent extraction and NMR sample preparation.

### 3.3. Optimization of Extraction Conditions

In order to improve the overall detection coverage of various types of metabolites of SR, the integral of 1D ^1^H NMR spectra of the extracts was used to evaluate optimized extraction conditions. Ultrasound assisted extraction was performed with 20%, 40%, 60%, 80%, 100% *v*/*v* methanol/water solutions; solid (sample)/liquid (solvent volume) ratios of 1:20, 1:60, 1:100, 1:140, 1:180, and 1:200; and extraction of 100, 250, 300, 400, 500, 750, and 1000 mg of sample.

### 3.4. NMR Sample Preparation and Data Acquisition

NMR samples were prepared as follows: 250 mg of ground SR was weighed and extracted with 40 mL 60% methanol/water solutions (*v*/*v*) in a 100 mL Erlenmeyer flask by ultrasonication at room temperature for 1 h. After centrifugation (10 min, 3077 g), the obtained supernatant was collected and evaporated using a rotary vacuum evaporator at 45 °C. Then, 600 µL of deuterated solvents (360 µL CD_3_OD + 240 µL D_2_O + 0.36 mg TSP) was added to redissolve the dried extracts of SR. The mixtures were then ultrasonicated for 3 min at room temperature and vortexed for 1 min. Finally, after centrifugation (10 min, 13,225 g), 500 µL of supernatant was transferred into a 5 mm NMR tube for NMR measurement.

For each sample, 1D ^1^H and 2D ^1^H-^13^C HSQC spectra were recorded at 298 K on a Bruker Avance III spectrometer (operating at 700.2034 MHz for ^1^H and 176.0795 MHz for ^13^C, respectively) equipped with a 5 mm QCI-F CryoProbe. The 1D ^1^H NMR spectra were acquired with a transmitter frequency of 4.798 ppm, relaxation delay of 2 s, 64 scans, 8 dummy scans, and a spectral width of 16 ppm. The 2D ^1^H-^13^C HSQC spectra were acquired with a 1.5 s relaxation delay; 8 transients; 512 increments; spectral widths of 12.98 and 150.00 ppm in F2 and F1, respectively; and offset 4.798 and 80 ppm for the ^1^H and ^13^C dimension, respectively.

### 3.5. NMR Spectrum Processing and Data Preprocessing

Two in-house TopSpin AU programs were developed to automate the batch processing of 1D ^1^H NMR spectra for all samples. One was for automatic Fourier transformation (EFP), zeroth order phase adjustment (APK0), baseline correction (ABS), and chemical shift calibration (SREF) of all 1D ^1^H NMR spectra. Considering the variation of the overall biomass among samples, the other AU program was developed to automate the segmented integration of all 1D ^1^H NMR spectra for the regions of 12.0–4.90, 4.78–3.38, and 3.30–0.05 ppm with the solvent peaks of water and residual proton of methanol-d4 excluded, and the integral of each ^1^H NMR spectrum was used as the normalization factor for the processing of the corresponding 2D ^1^H-^13^C HSQC spectra.

An in-house NMRPipe script was used to automate the processing of all 2D ^1^H-^13^C HSQC spectra in-batch. To obtain unbiased and comparatively meaningful absolute intensities, the scaling parameter (NC) was read and applied to 2D ^1^H-^13^C HSQC raw data besides normalization by the integral of the corresponding ^1^H NMR spectrum.

Peaks were identified and quantified in all 2D ^1^H-^13^C HSQC spectra using a Java-based program, Newton, which employed an algorithm called fast maximum likelihood reconstruction (FMLR) for 2D ^1^H-^13^C HSQC spectral deconvolution to extract peaks and to obtain accurate signal quantitation [31]. Newton supports batch analysis of a series of related datasets, and all processed 2D ^1^H-^13^C HSQC spectra can be imported into Newton together. An appropriate threshold was set, and a prototype peak was selected as model peak for peak fitting. The batched spectral peaks obtained from Newton analysis were then preliminarily matched, forming peak clusters across samples using an in-house developed R program (manuscript in preparation). After comparing and uniquely designating the ambiguous peaks to a specific peak cluster and merging multiple redundant peaks from the same sample in the same peak cluster, the obtained peak clusters were further filtered to generate the valid peak clusters (spectral features). The valid peak clusters were then refilled. Finally, the regularized matrix of peak intensity (*M*) with the different features and samples as columns and rows, respectively, was obtained for the following chemometric analysis. Excluding the signals from the reference TSP and the residual protonated solvents, the final data matrix of peak intensity has 474 features and 99 measurements (that is, number of samples) for each feature; that is, the size of the matrix is 99 × 474.

### 3.6. Chemometric Analysis

All data were mean centered and scaled to unit variance (UV) as a preprocessing step then analyzed with principal component analysis (PCA) and orthogonal partial least squares discriminant analysis (OPLS-DA) or partial least squares discriminant analysis (PLS-DA) based on the matched and aligned spectral peaks obtained by Newton analysis, using SIMCA-P (version 14.1, Umetrics, Umea, Sweden). Seven-fold internal cross-validation was also performed to verify these models [32,33,34]. The quality of the OPLS-DA and PLS-DA models was revealed by the parameters of R^2^ and Q^2^ [32,35]. R^2^ indicates the goodness of fit, and Q^2^ represents the predictive power of the model. The values of R^2^ and Q^2^ close to 1.0 indicate an excellent model [32,35]. The values of variable importance for projection (VIP) and *p*(corr) were used to evaluate the contribution or importance and the relevance of individual X-variables to the OPLS-DA or PLS-DA model [27,36]. The variables with a VIP value higher than 1 and high *p*(corr) correlation value are considered as potential discriminant variables associated with the classification [36,37].

### 3.7. Metabolite Identification and Verification

We previously developed an automatic approach, NPid to rapid identification of known natural products in a mixture of extracts based on a 2D ^1^H-^13^C HSQC spectrum [38]. For NPid analysis, a customized NMR database of SR was constructed by using the chemical shifts and structural information of natural products in SR retrieved from published phytochemical research. In 2D ^1^H-^13^C HSQC spectra, each chemical constituent could be characterized by multiple well-resolved ^1^H-^13^C cross-peaks.

The differential features selected were analyzed using NPid and preliminarily assigned by comparing with the customized NMR database of SR obtained through literature data mining and public spectral databases (HMDB, BMRB). Each metabolite was identified and represented by multiple resolved ^1^H-^13^C cross-peaks, and each ^1^H-^13^C HSQC cross-peak contained paired chemical shifts of ^1^H and ^13^C. When standard compounds were available, the preliminarily identified metabolites were further verified by comparing the 2D ^1^H-^13^C HSQC spectra of the mixture with the standards of the candidate metabolites.

## 4. Conclusions

In this study, a 2D ^1^H-^13^C HSQC-based untargeted metabolomics approach was successfully applied to extensively characterize the difference in chemical components of SR processed by steaming for different durations and using different processing methods. These findings could elucidate the change of chemical constituents of the processed SR and provide a guide for the processing. Comprehensive characterization of the effect of the processing methods on the chemical constituents of TCM is a premise for follow-up research, such as exploring the therapeutic basis and establishing quality control standards and disclosing the underlying secret of the effectiveness and mechanisms of TCM processing.

## Figures and Tables

**Figure 1 molecules-27-04687-f001:**
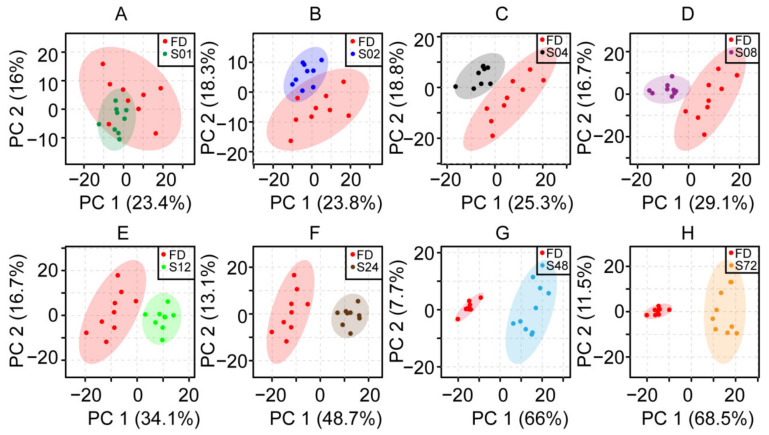
The PCA score plots of 2D ^1^H-^13^C HSQC spectra of SR steamed for different durations. Comparison between the vacuum freeze-dried (FD, red) samples and samples steamed for (**A**) 1 (S01, green), (**B**) 2 (S02, blue), (**C**) 4 (S04, black), (**D**) 8 (S08, purple), (**E**) 12 (S12, light green), (**F**) 24 (S24, brown), (**G**) 48 (S48, light blue), and (**H**) 72 h (S72, yellow), respectively.

**Figure 2 molecules-27-04687-f002:**
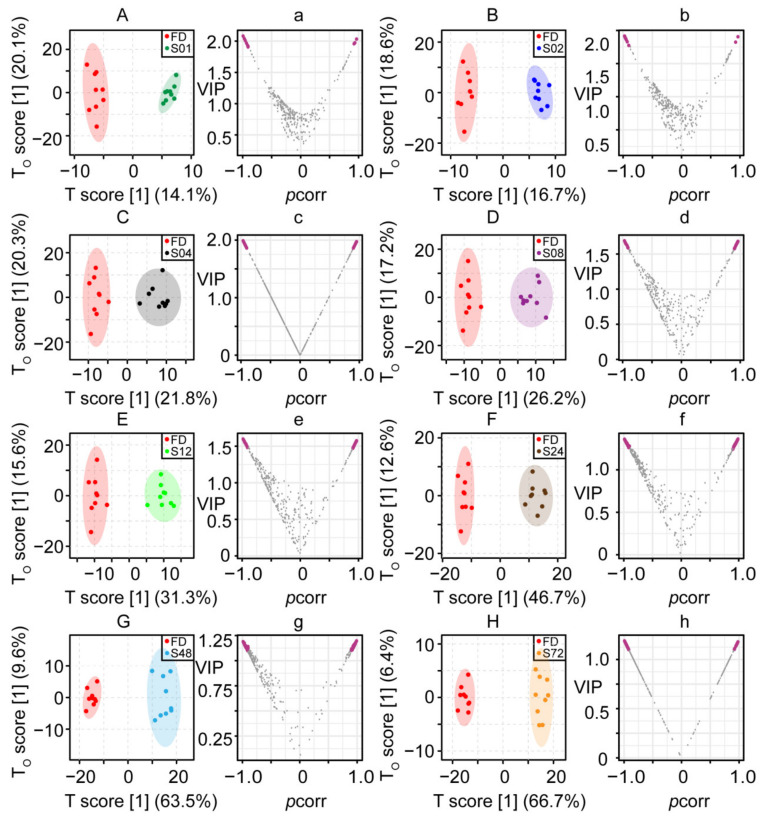
The OPLS-DA score plots of 2D ^1^H-^13^C HSQC spectra of SR steamed for different durations. Comparison between vacuum freeze-dried (FD, red) samples and samples steamed for (**A**) 1 (S01, green), (**B**) 2 (S02, blue), (**C**) 4 (S04, black), (**D**) 8 (S08, purple), (**E**) 12 (S12, light green), (**F**) 24 (S24, brown), (**G**) 48 (S48, light blue), and (**H**) 72 h (S72, yellow). The volcano plots of VIP and *p*(corr) of variables from OPLS-DA between vacuum freeze-dried (FD) samples and samples (**a**) S01, (**b**) S02, (**c**) S04, (**d**) S08, (**e**) S12, (**f**) S24, (**g**) S48, (**h**) S72. Differential variables with VIP > 1.1 and |*p*(corr)| > 0.9 are highlighted in magenta.

**Figure 3 molecules-27-04687-f003:**
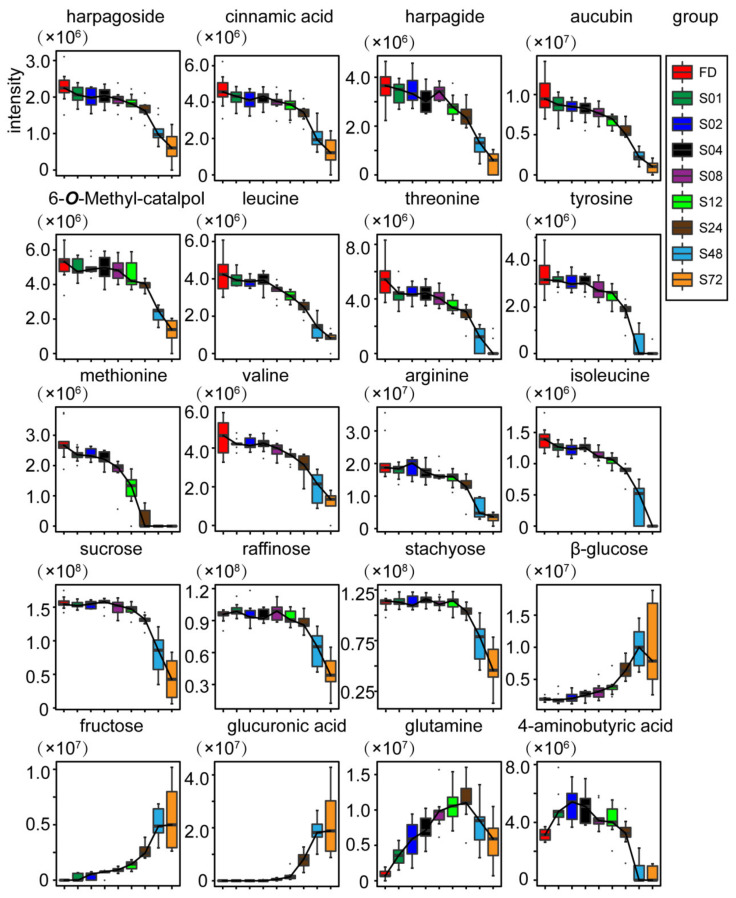
Boxplots of the relative intensities of 20 differential compounds at different steaming durations, 1 (S01, green), 2 (S02, blue), 4 (S04, black), 8 (S08, purple), 12 (S12, light green), 24 (S24, brown), 48 (S48, light blue), and 72 h (S72, yellow), compared with the vacuum freeze-dried (FD, red) samples.

**Figure 4 molecules-27-04687-f004:**
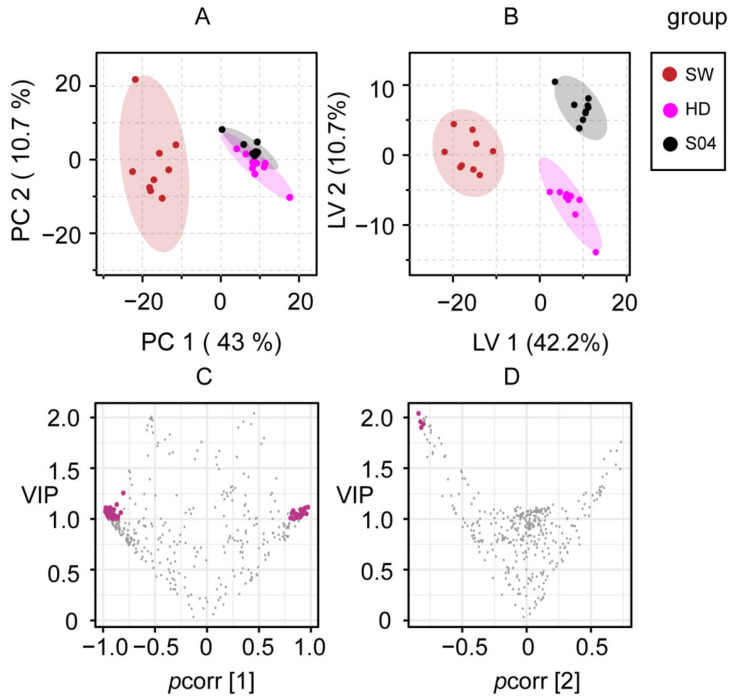
Multivariate analysis of 2D ^1^H-^13^C HSQC spectra of SR processed using different methods, sweating (SW, claret), hot-air drying (HD, peach), and steaming for 4 h (S04, black). (**A**) PCA and (**B**) PLS-DA score plots. (**C**,**D**) Volcano plots of VIP and *p*(corr) of variables from PLS-DA. Differential variables with VIP > 1.0 and |*p*(corr)| > 0.8 are highlighted in magenta.

**Figure 5 molecules-27-04687-f005:**
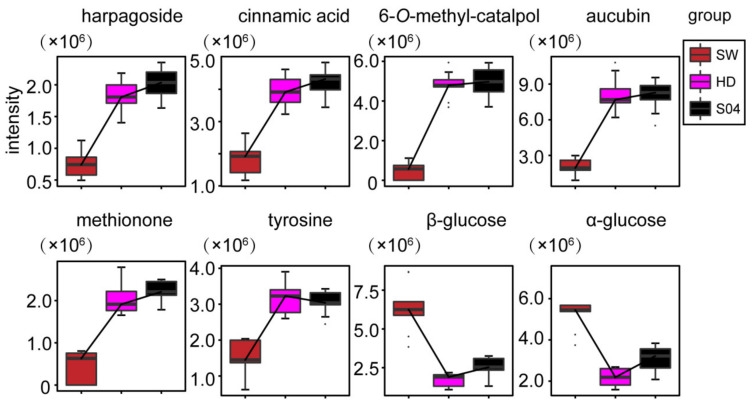
Boxplots of the relative intensities of seven identified differential compounds in samples processed using different methods, sweating (SW, claret), hot-air drying (HD, peach), and steaming for 4 h (S04, black).

## Data Availability

Compiled data is reported in the tables above. The raw data files are available from authors upon request.

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
