# Peer review of "Effect of Different Processing Methods on the Chemical Constituents of Scrophulariae Radix as Revealed by 2D NMR-Based Metabolomics"

_molecules, 2022, doi:10.3390/molecules27154687_

Round 1

Reviewer 1 Report

The manuscript of  Duan et al. reports the chemometric study of one of the popular Chinese medicinal plants Scrophulariae Radix by heteronuclear 2D NMR. The influence of the processing method on the chemical composition of the extract  is accessed by several statistic methods. The difference between sweating, freeze-drying, hot-air-drying and steaming are apparent from PCA and OPLS-DA plots. Most of the compounds responsible for the variance are identified. The experimental data and their interpretations are solid. This paper presents a nice example of research in the field of NMR-based metabolomics and should present an interest for the readership of Molecules. 

I recommend an acceptance of this paper after correcting a couple of minor issues:

1)    Authors claim that the decrease of the amino acid content during a prolonged steaming may be due to Mallard reaction. Is it possible to identify some of the common products of this reaction such as Amadori products or advanced glycation endproducts

2)    Did authors recalibrate the pulse width for each sample during NMR measurements, or the same value was used for all experiments?

3)    Line 286, end of the sentence, add the missing word.

4)    Line 167, a line break is missing. 

Reviewer 2 Report

I enjoyed very much to read this paper and I think this approach is very inetresting and relevant to the field. qNMR and particularly 2D qNMR is indeed a valuable technique for metabolomics and chemometric analysis.

My only question is about the choice of solvent: why methanol which is not a GRAS solvent

Reviewer 3 Report

The article is very interesting and has many applications.

The use of 2DNMR data in chemometrics seems very useful.

I would advise its publication after revision.

The statistical models have to be validated through permutation testing (999 permutations) and ROC Curves, otherwise we cannot  be sure if they are reliable and not overfitted. This is a necessary step that cannot be ommitted.

Moreover, more information regarding the processing of the 2D NMR data prior to the statistical analysis must be given in the materials and methods section. I would advise the authors to also refer to pitfalls and important steps that need highlighting during this process.
